# Frozen Section of Placental Membranes and Umbilical Cord: A Valid Diagnostic Tool for Early-Onset Neonatal Sepsis Management

**DOI:** 10.3390/diagnostics14111157

**Published:** 2024-05-31

**Authors:** Veronica Parrella, Michele Paudice, Michela Pittaluga, Alessandra Allodi, Ezio Fulcheri, Francesca Buffelli, Fabio Barra, Simone Ferrero, Cesare Arioni, Valerio Gaetano Vellone

**Affiliations:** 1Department of Integrated Surgical and Diagnostic Sciences (DISC), University of Genoa, Viale Benedetto XV, 14, 16132 Genoa, Italy; veronicaparrella19@gmail.com (V.P.); michele.paudice@unige.it (M.P.); ezio.fulcheri@fastwebnet.it (E.F.); 2Pathology University Unit, IRCCS Ospedale Policlinico San Martino, Largo R.Benzi, 10, 16132 Genoa, Italy; 3Department of Internal Medicine (DIMI), University of Genoa, Viale Benedetto XV, 14, 16132 Genoa, Italy; michelapittaluga11@gmail.com; 4Neonatology Unit, IRCCS Ospedale Policlinico San Martino, Largo R.Benzi, 10, 16132 Genoa, Italy; alessandra.allodi@hsanmartino.it (A.A.); cesare.arioni@hsanmartino.it (C.A.); 5Pathology Unit, IRCCS Istituto Giannina Gaslini, Via G.Gaslini, 5, 16147 Genoa, Italy; francescabuffelli@gaslini.org; 6Obstetrics and Gynecology Unit, ASL4, Via Don Giovanni Battista Bobbio, 25, 16033 Lavagna, Italy; fabio.barra@icloud.com; 7Obstetrics and Gynecology University Unit, IRCCS Ospedale Policlinico San Martino, Largo R.Benzi, 10, 16132 Genoa, Italy; simone.ferrero@me.com; 8Department of Neurosciences, Rehabilitation, Ophthalmology, Genetics, Maternal and Child Health (DINOGMI), University of Genoa, L.go Paolo Daneo 3, 16132 Genoa, Italy

**Keywords:** neonatal sepsis, placenta, funisitis, chorioamnionitis, frozen section examination

## Abstract

Early-onset neonatal sepsis (EONS), a serious infection in newborns within 3 days, is challenging to diagnose. The current methods often lack accuracy, leading to unnecessary antibiotics or delayed treatment. This study investigates the role of the frozen section examination of placental membranes and umbilical cord (FSMU) to improve EONS diagnosis in the daily lab practice. This retrospective study reviewed data from 59 neonates with EONS risk factors who underwent FSMU according to our institutional protocol. Concordance between the FSMU and the Final Pathological Report (FPR) was assessed. The FSMU demonstrated a high concordance (Kappa = 0.88) for funisitis diagnosis, with excellent accuracy (98.3%). A moderate concordance was observed for chorioamnionitis stage and grade. The FSMU shows promise as a rapid and accurate tool for diagnosing EONS, particularly for funisitis. This study suggests that the FSMU could be a valuable tool for EONS diagnosis, enabling a more judicious antibiotic use and potentially improving outcomes for newborns.

## 1. Introduction

Early-onset neonatal sepsis (EONS), a systemic infection affecting newborns within the first three days of life, represents a significant contributor to morbidity and mortality [1].

Among maternal risk factors, Group B Streptococcus (GBS) colonization represents the leading cause of EONS, particularly in developed countries. Maternal colonization rates can be as high as 35% [2,3]. A routine GBS screening during pregnancy allows for antibiotic prophylaxis during labor, significantly reducing the risk of EONS transmission to the newborn [3]. Indeed, the inflammatory condition of the fetal membranes (chorioamnionitis) is often caused by ascending bacterial infections with maternal symptoms, like fever, uterine tenderness, and foul-smelling amniotic fluid. Furthermore, when the amniotic sac ruptures prematurely, often before labor begins, configuring the condition defined as prolonged rupture of membranes (PROMs), it exposes the fetus to potential pathogens for a longer duration, increasing the risk of EONS.

Premature infants, especially those with very low birth weights, have immature immune systems, making them more susceptible to infections like EONS. The presence of meconium in the amniotic fluid can increase the risk of EONS by facilitating the passage of bacteria into the bloodstream during aspiration.

Newborns facing delivery complications, like birth asphyxia and low Apgar scores, have weakened immune systems, heightening vulnerability to EONS. Additionally, newborns needing invasive procedures, like umbilical catheterization, face increased EONS risk due to potential bacterial introduction into the bloodstream [4].

Newborns with EONS may present with a wide range of signs and symptoms, making diagnosis a multifaceted task. Some neonates may exhibit fever or hypothermia. However, temperature instability is a common finding in healthy newborns, reducing its diagnostic value in isolation.

Feeding difficulties, respiratory distress like rapid breathing or grunting, and gastrointestinal symptoms, such as vomiting or diarrhea, can indicate EONS. Neurological abnormalities, like seizures or irritability, may also be present but require careful evaluation to distinguish from other causes. Additionally, non-specific signs, like lethargy, jaundice, or poor blood flow, can contribute to the diagnostic challenge of EONS [1,5,6].

Broad-spectrum antibiotic therapy is frequently employed as a precautionary measure due to the difficulty in confirming or excluding sepsis. This practice contributes to the growing problem of antibiotic resistance. On the contrary, inaccurate or slow diagnostic tests can delay the initiation of appropriate, targeted therapy, potentially impacting the psycho-physical development of the affected neonate [7,8].

The conventional diagnostic tools for EONS, such as complete blood count (CBC) and blood cultures, often lack specificity and sensitivity. Furthermore, blood cultures can take hours to days to yield results, significantly delaying the initiation of an appropriate treatment [9,10,11,12].

Several laboratory tests measure markers of inflammation, such as C-reactive protein (CRP) and white blood cell (WBC) count. While these can be elevated in EONS, they are also non-specific and can be increased in response to prematurity or birth asphyxia [13].

While antibiotics are undoubtedly lifesaving in confirmed cases of sepsis, broad-spectrum antibiotic use can disrupt the newborn’s developing gut microbiome, increasing susceptibility to future infections and immune-related conditions [14,15,16,17].

Frozen section examination, a rapid histological technique commonly used in oncological surgery, offers a promising approach for diagnosing EONS. This technique involves quickly freezing a tissue sample, preparing a thin section, and staining it to allow for immediate microscopic evaluation [18,19]. In the context of EONS, a frozen section of placental membranes and umbilical cord (FSMU), especially looking for funisitis (fetal inflammatory response), could rapidly guide clinicians regarding the presence and severity of an inflammatory response [20,21].

Therefore, in this study, we conducted a retrospective evaluation of our case series to gauge the reliability and limitations of FSMU, juxtaposing it with the final pathological report (FPR).

## 2. Materials and Methods

### 2.1. Setting and Population Inclusion Criteria

This retrospective study was conducted at the Neonatology Unit of the Gynecological and Birth Pathway Department, San Martino Polyclinic Hospital, Italy. The unit provides care for newborns with a gestational age greater than 32 weeks. The study included all consecutive neonates born between 2019 and 2021 who underwent FSMU as part of their sepsis evaluation.

The risk factors, to be intended as indications for FSMU were those presented in Table 1 [22,23].

### 2.2. Data Sources

This study utilized data from three primary sources:

FSMU reports: These reports documented the indications for FSMU, the presence of funisitis, and the presence, stage, and grade of chorioamnionitis. Indications were categorized as absolute (meeting specific criteria) or relative (suggestive of potential risk).

FPR: This report detailed the macroscopic and microscopic findings (on formalin-fixed and paraffin-embedded (FFPE) tissue) of the umbilical cord, membranes, and placental chorial disc, following an institutional protocol based on the 2016 Amsterdam Consensus Conference [24].

Neonatal Medical Records: Electronic medical records from the TrakCare program™ were reviewed to gather data on maternal and fetal conditions.

### 2.3. FSMU Technique

The frozen section examination was performed on four tissue samples: three sections from different levels of the umbilical cord (the middle, the point near the insertion to the placental disc, and the opposite point toward the fetus) and one section of the membranes (“membrane roll”, cutting a strip from the rupture site to the placental insertion). Freshly collected samples were snap-frozen at −20 °C. For each sample, two 5 μm thick histological sections were cut using a freezing microtome and stained with hematoxylin and eosin (H&E) and toluidine blue (Figure 1).

In the same session, three chorionic disc samples were taken (within the central two-thirds of the disc, with one encompassing the umbilical cord insertion area) for the subsequent FPR. The two frozen blocks utilized for the FSMU analysis were then thawed for the FPR, which included fixation in 10% buffered formalin for 12 h, processing, paraffin embedding (FFPE), sectioning at 3 μm thickness, and staining with H&E (Figure 2).

### 2.4. Data Analysis

Data on FSMU indications, funisitis, and chorioamnionitis details were extracted from the FSMU reports. Data on FSMU reports were compared with those of the FPR. All information, including pathological and clinical data, was entered into a Microsoft Excel™ Version 14.0.7628.5000 spreadsheet. Statistical analysis was performed using the MedCalc™ Version 22. 023 software.

## 3. Results

This study investigated the concordance between the FSMU and the FPR in diagnosing funisitis and chorioamnionitis in a cohort of 59 neonates with risk factors for early-onset sepsis (EONS).

### 3.1. Patient Demographics

A total of 59 consecutive infants were included (mean gestational age: 38 weeks). The characteristics of the study population are illustrated in Table 2.

The mean maternal age was 32.4 years, with a majority being primiparae (57.63%).

### 3.2. Indications for FSMU

Most patients (43 cases; 72.88%) had absolute indications for FSMU, with PROMs being the most frequent (59.32%).

Relative indications (16 cases; 27.12%) included meconium-stained amniotic fluid, pre-eclampsia, gestational diabetes, and cardiotocographic changes (CTGs). Each considered patient could have more than one indication(s), including both absolute and relative.

### 3.3. FSMU and FPR Concordance

The results of the FSMU and the FPR for funisitis and chorioamnionitis and their concordance are illustrated in Table 3.

### 3.4. Clinical Outcomes and Antibiotic Use

The clinical records revealed the following data:

Most newborns (*n* = 42; 72.88%) did not require antibiotics and were discharged.

Sixteen newborns (27.12%) required empiric antibiotic therapy until sepsis was clinically ruled out (clinical improvement). In 10 of these cases, antibiotic therapy was initiated due to the presence of at-risk neonatal symptoms (all with respiratory distress), regardless of the FSMU results. In the other six cases, although the newborns were asymptomatic, the antibiotic therapy was started because of funisitis, detected in the FSMU (four cases), and severe leukocytosis (white cell count of 32.480/mmC) (one case). These infants were eventually discharged in good health with no complications at follow-up. One infant with a congenital heart defect was transferred to a higher-level pediatric hospital (IRCCS Istituto Giannina Gaslini) due to complications.

## 4. Discussion

The incidence of EOS in Italy, estimated at 0.61/1000 live births, has decreased thanks to the screening for GBS at 35–37 weeks of gestation and intrapartum antibiotic prophylaxis [25].

The early recognition and management of EONS are crucial for improving neonatal outcomes. By understanding the various risk factors, healthcare professionals can implement preventive measures, like GBS screening and antibiotic prophylaxis, during labor. Additionally, the close monitoring of newborns at risk can facilitate early diagnosis and prompt treatment with antibiotics, potentially preventing serious complications.

Clinical laboratory tests are valuable tools in the diagnosis of EONS, but their limitations must be recognized. The slow turnaround time of blood cultures and the non-specific nature of inflammatory markers necessitate a cautious approach.

The current diagnostic challenges in EONS often lead to overtreatment with broad-spectrum antibiotics. These antibiotics, while essential for confirmed sepsis cases, can have devastating consequences for newborns. They indiscriminately kill both harmful and beneficial bacteria in the developing gut microbiome. This disruption can have long-lasting effects on the immune system, increasing susceptibility to future infections and other immune-modulated impairments [14,15,16,17].

This study highlights the role of the FSMU as a valid additional diagnostic to aid in the effective management of newborns at risk of neonatal sepsis. By rapidly and accurately identifying inflammation in membranes (a marker of maternal inflammatory response) and especially in the umbilical cord (a marker of fetal inflammatory response) [20], the FSMU can help clinicians to decide whether or not to administer antibiotics, minimizing unnecessary exposure.

To the best of our knowledge, the only other work in the literature that specifically considered the accuracy of the FSMU was by Mahe et al. [26].

The FSMU involves detecting neutrophilic granulocytes in the membranes and umbilical cord. In theory, this examination does not require significant diagnostic efforts. This is certainly an advantage, as it can be easily applied outside specialized centers. A brief training and knowledge of histological criteria [24] for defining chorioamnionitis and funisitis are sufficient, even for a pathologist not specifically dedicated to fetal–placental pathology.

However, like every diagnostic test, the FSMU is not foolproof and can yield false positives or false negative results. In our case series, we did not identify any false negatives for funisitis. In essence, among the 59 cases of newborns potentially at risk of early-onset neonatal sepsis (EONS), the frozen section examination did not miss any instances of funisitis when compared to the FPR.

Unfortunately, one of the issues that can affect frozen section examination is freezing artifacts and artifacts resulting from cutting frozen tissue. The only case of discordance for funisitis between the FSMU and the FPR resulted from a pathologist’s evaluation error caused by freezing artifacts. This case also showed chorioamnionitis (confirmed in the final report). The constriction of the frozen umbilical vein wall led to the misinterpretation of focal funisitis. However, while emphasizing the importance of integrating with other clinical–laboratory data, in this case, the newborn would still have been indicated to start antibiotic therapy even in the absence of funisitis, as the newborn presented multiple risk factors (PROMs > 18 h, meconium-stained amniotic fluid, and chorioamnionitis) and symptoms (severe hypotonia at birth).

Excluding this case, all four confirmed cases of funisitis were stage 1 (involving only the umbilical vein) and grade 1 (non-severe), and all were associated with chorioamnionitis (stage II). Of these, three cases were related to newborns who were completely asymptomatic at birth and with negative lab exams (negative blood culture). This finding, even though we are aware of the limited representativeness of these small numbers, appears interesting because the presence of funisitis detected in the FSMU helped the clinician in the decision to start antibiotic therapy.

Regarding inflammation in the membranes, it is noted that, among 43 cases where inflammation was not evident in the FSMU, 5 cases revealed focal stage I chorioamnionitis (confined to the chorion), grade 1 on the FPR. This highlights that, for highly focal and minimal lesions (with sparse inflammatory infiltrate), inflammation might not be detected in sections prepared for the FSMU but could appear in subsequent sections for the FPR, likely with little consequence to the newborn’s health.

It is nonetheless important to emphasize the additional role of the FSMU compared to others available to the clinician. The FSMU should not be seen as a substitute but as an adjunctive device that can be quickly performed in the clinical practice.

In this regard, when there is strong clinical suspicion for EONS, the neonatologist may initiate antibiotic therapy even before the FSMU results are available. In our case series, for instance, 10 symptomatic newborns with respiratory distress received antibiotic therapy even in the absence of funisitis on the FSMU. In these cases, the FSMU still holds significance for the neonatologist as it allows for the modulation of antibiotic therapy, whether to escalate or de-escalate it.

This study acknowledges areas for further exploration. Multicenter studies and meta-analyses with larger sample sizes are needed to solidify these findings. Additionally, refining the FSMU technique or interpretation could enhance its diagnostic accuracy.

While the upfront cost of implementing the FSMU may be a concern, future studies could analyze its cost-effectiveness. A reduced unnecessary antibiotic use translates to lower drug costs and potentially shorter hospital stays. Additionally, the potential long-term health benefits associated with a preserved microbiome could lead to further cost savings down the line.

## 5. Conclusions

This study offers strong evidence supporting the FSMU as a valuable tool for diagnosing EONS. The FSMU is a quick, sustainable, easily applicable, and interpretable test, even in pathology laboratories where a pathologist specialized in fetal–placental pathology is not available. By accurately identifying high-risk neonates, the FSMU empowers clinicians to administer targeted antibiotic therapy, minimizing unnecessary antibiotic exposure. Further research with a larger sample size is essential to solidify these findings and explore the broader role of the FSMU in diagnosing EONS. This advancement has the potential to pave the way for a future with healthier newborns and a more judicious approach to antibiotic use.

## Figures and Tables

**Figure 1 diagnostics-14-01157-f001:**
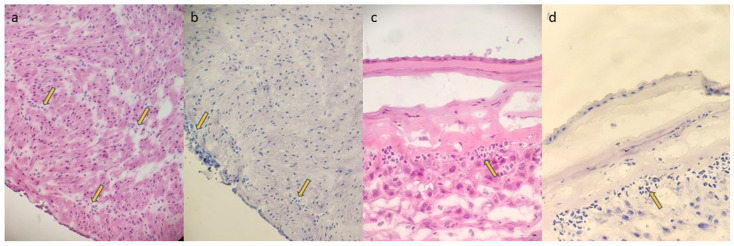
(FSMU; original magnification 200×): (**a**,**b**): funisitis; yellow arrows show neutrophilic granulocyte infiltrates in the wall of the umbilical vein ((**a**): H&E; (**b**): toluidine blue); (**c**,**d**): chorioamnionitis stage 2 grade 2; yellow arrows show neutrophilic granulocyte infiltrates in the chorion of the membranes and minimally in the sub- amniotic connective tissue ((**c**): H&E; (**d**): toluidine blue).

**Figure 2 diagnostics-14-01157-f002:**
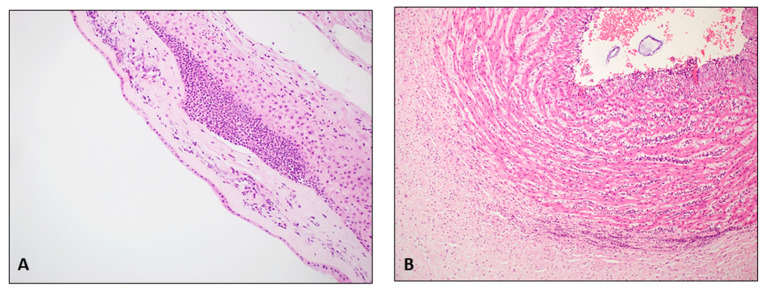
(FPR on FFPE tissue; hematoxylin/eosin; original magnification 200×): (**A**): severe chorioamnionitis G2 stage 2; widespread neutrophilic granulocyte infiltrates in the chorion with extension into the sub-amniotic connective tissue (**B**): severe funisitis; numerous neutrophilic granulocytes in the wall of the umbilical vein.

**Table 1 diagnostics-14-01157-t001:** Risk factors for EONS are intended as indications for FSMU.

Risk Factor	Description
Chorioamnionitis or peripartum maternal fever	- Histological evidence of chorioamnionitis on placental pathology OR - Maternal temperature > 38 °C within 48 h before or after delivery
Prolonged rupture of membranes (PROMs)	Rupture of membranes for ≥18 h associated with: - No antibiotic therapy; - Incomplete antibiotic therapy; - Inadequate antibiotic therapy.
Preterm premature rupture of membranes (PPROMs)	Rupture of membranes before 37 weeks of gestation, particularly unexplained cases
Positive vagino-rectal swab for fetal pathogens	Positive swab for Group B Streptococcus (GBS) or other pathogens with: - No intrapartum antibiotic prophylaxis; - Incomplete intrapartum antibiotic prophylaxis; - Inadequate intrapartum antibiotic prophylaxis.
Maternal genitourinary infection during pregnancy	Documented maternal genitourinary infection during pregnancy with: - No antibiotic treatment OR; - No negative control swab following treatment.
Co-presence of multiple risk factors	Two or more of the listed risk factors are present
Emerging infectious risk factors	Any other potential infectious risk factors identified on an individual basis by the neonatologist

**Table 2 diagnostics-14-01157-t002:** Characteristics of the study population.

Characteristic	Mean	Min–Max
Maternal age	32.42	22–43
Gestational age	38.02	32–42
Birth weight	2.915	1440–4060
Characteristic	Number	Percentage (%)
Gravida		
G1	24	40.68
G2	19	32.20
>G3	16	27.12
Para		
P1	34	57.63
P2	16	27.12
>P3	9	15.25
Sex of the Newborn		
Female	30	50.85
Male	29	49.15
Apgar Score 1′		
0–2	0	0.00
3–5	3	5.08
6–8	15	25.42
9–10	39	66.10
N/A	2	3.39
Apgar Score 5′		
0–2	0	0.00
3–5	0	0.00
6–8	9	15.25
9–10	48	81.36
N/A	2	3.39
**Absolute Indication**		
Premature rupture of membranes (PROMs)	35	59.32
Vaginal and/or rectal swabs positive for GBS	12	20.34
Vaginal and/or rectal swabs for GBS not performed	6	10.17
Maternal fever	2	3.39
**Relative Indication**		
Tinted or malodorous amniotic fluid	10	16.95
Pre-eclampsia or gestational diabetes mellitus	6	10.17

**Table 3 diagnostics-14-01157-t003:** Concordance between the FSMU and the FPR.

FUNISITIS Frozen Sections		FUNISITIS FPR		
	NO	Stage 1	Stage 2	TOTAL
NO	54	0	0	54 (91.5%)
YES	1	3	1	5 (8.5%)
TOTAL	55 (93.2%)	3 (5.1%)	1 (1.7%)	59 (100%)
SIGNIFICANCE LEVEL				*p* < 0.0001
95% CI				0.64801 to 1
**CHORIONAMIONITIS STAGE** **Frozen Sections**		**CHORIONAMNONTIS STAGE** **FPR**		
	NO	Stage 1	Stage 2	TOTAL
NO	38	5	0	43 (72.9%)
Stage 1	4	7	0	11 (18.6%)
Stage 2	0	1	4	5 (8.5%)
TOTAL	42 (71.2%)	13 (22%)	4 (6.8%)	59 (100%)
SIGNIFICANCE LEVEL				*p* < 0.0001
95% CI				0.49696 to 0.88002
**CHORIONAMIONITIS GRADE** **Frozen Sections**		**CHORIONAMNONTIS GRADE** **FPR**		
	NO	G1	G2	TOTAL
NO	37	4	0	41
G1	5	9	1	15
G2	0	2	1	3
TOTAL	42 (71.2%)	15 (25.4%)	2 (3.4%)	59 (100%)
SIGNIFICANCE LEVEL				*p* < 0.0001
95% CI				0.33121 to 0.77517

Funisitis: A high concordance (Kappa coefficient, K = 0.88) was observed. FSMU demonstrated excellent accuracy (98.3%, 95% CI: 90.91–99.96%) with high sensitivity (100%, 95% CI: 39.76–100%) and specificity (98.18%, 95% CI: 90.28–99.95%). Only one case of funisitis identified by the FSMU was not confirmed by the FPR. Chorioamnionitis: A moderate concordance was observed for both the stage (K = 0.55) and grade (K = 0.69) of chorioamnionitis.

## Data Availability

Raw data of the submitted study are available by contacting the corresponding author.

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
