# Peer review of "Frozen Section of Placental Membranes and Umbilical Cord: A Valid Diagnostic Tool for Early-Onset Neonatal Sepsis Management"

_diagnostics, 2024, doi:10.3390/diagnostics14111157_

Round 1
Reviewer 1 Report
Comments and Suggestions for Authors
Thank you for offering me the opportunity to review this manuscript.
The study is promising and certainly brings novelty to the scientific community.
Unfortunately, the sample size is small and cannot provide sufficient strength to the statistics. It is certain that further analysis could bring more insights into the field and future research (on more samples) could provide more publishing material.
Nevertheless, some issues need addressing and clarification in order to provide more scientific soundness and to offer insights to the scientific community.
- Line 159 - the PFSE, FFPE techniques – it is not clear how many sections were used etc. please elaborate.
- Is there a dedicated pathologist? What experience do the pathologists have in the field?
- Line 220 – which antibiotics were used? Define long course of atb.
- line 221 – negative blood culture or negativization after a positive test
- line 231 – how many patients included in the study did not receive a swab for BGS or did not receive correct intrapartum antibiotic prophylaxis? How many of those newborn have developed a septic condition?
- Please state more clearly how the histopathological result could influence the clinical practice. When no microscopically proof of infection was noted was the antibiotic avoided, even if the neonatologist would indicate it?
- If it were to be implemented in the clinical practice, do you consider the costs reasonable? Financial cost and also personnel etc. especially compared to extensive cultures, blood cultures, different culture media etc.
Author Response
Reviewer: 1
Thank you for offering me the opportunity to review this manuscript.
The study is promising and certainly brings novelty to the scientific community.
Unfortunately, the sample size is small and cannot provide sufficient strength to the statistics. It is certain that further analysis could bring more insights into the field and future research (on more samples) could provide more publishing material.
Nevertheless, some issues need addressing and clarification in order to provide more scientific soundness and to offer insights to the scientific community.
- Line 159 - the PFSE, FFPE techniques – it is not clear how many sections were used etc. please elaborate.
In line 158 we described the frozen section technique for PFSE: “three sections from different levels of the umbilical cord and one section of the membranes. Freshly collected samples were snap-frozen at -20°C. For each sample, two 5 μm thick histological sections were cut using a freezing microtome and stained with hematoxylin and eosin (H&E) and toluidine blue”.
We specify and add in the text the points in which we sampled the umbilical cord (the middle, the point just near the insertion to the placental disc, and the opposite distal point). Moreover, we specify the method of “membrane roll” for the examination of membranes, cutting a strip from the rupture site to the placental insertion.
- Is there a dedicated pathologist? What experience do the pathologists have in the field?
Yes, there are pathologists dedicated to fetal-placental pathology. However, the intraoperative consultation is interdisciplinary, and for this reason, all of our pathologists have undergone appropriate training for the proper management of PFSE.
It's also necessary to consider the relative ease of PFSE even for those not specializing in feto-placental pathology. This is because any pathologist, after studying the staging and grading system for chorioamnionitis and funisitis, can easily recognize the presence of neutrophil granulocytes indicative of inflammation.
This is certainly an advantage for the introduction of such a method, as it could easily be implemented even in more peripheral centers where subspecialization is less guaranteed.
- Line 220 – which antibiotics were used? Define long course of atb.
The study aims to demonstrate the usefulness of PFSE in identifying neonatal sepsis quickly and accurately. Therefore, we haven't delved into the details of antibiotic therapies. However, the therapy of choice consists of a combination of Ampicillin + Gentamicin. In case of clinical worsening after 36 hours of therapy and/or meningitis, a combination with a third-generation cephalosporin (Cefotaxime or Ceftazidime) is recommended. However, their use remains limited as they promote the emergence of multi-resistant Gram-negative strains and systemic infections by Candida. Antibiotic therapy should be discontinued when blood cultures are sterile at 48-72 hours of incubation unless the clinical presentation is particularly suggestive or there is a site-specific infection. In the case of confirmed sepsis, treatment should be continued for 10 days, 14 days if there is GBS bacteremia and 21-28 days in the case of meningitis. Determination of blood levels of aminoglycosides should be performed when therapy with these drugs is continued for more than 72 hours or in the presence of renal insufficiency: if the levels remain below 2 mcg/ml, the risk of ototoxicity and nephrotoxicity is significantly reduced.
- line 221 – negative blood culture or negativization after a positive test
The case in question had a negative blood culture since the beginning.
Furthermore, it is unfortunately well known that blood cultures can yield false negative results.
- line 231 – how many patients included in the study did not receive a swab for BGS or did not receive correct intrapartum antibiotic prophylaxis? How many of those newborn have developed a septic condition?
Generally, all pregnant women are recommended to undergo a swab for BGS around the 35th week. In our case series, all except six women (as we wrote in Table 2) had undergone the test; in one case the reason was due to preterm delivery (at 31 weeks).
Moreover, twelve cases were positive for BGS.
However, no cases of Neonatal Sepsis requesting intensive care were found.
- Please state more clearly how the histopathological result could influence the clinical practice. When no microscopically proof of infection was noted was the antibiotic avoided, even if the neonatologist would indicate it?
As we indicated in the text, the PFSE is a tool that, with low costs and quick turnaround times, can guide the clinical colleague's choices. Therefore, it represents a complementary examination and not a complete substitute for other clinical investigations.
Furthermore, it can also retrospectively guide the clinician in terms of modulating therapy in cases where antibiotic therapy has already been initiated due to a high clinical suspicion of infection.
It should also be considered that PFSE is not infallible since only a small portion of the entire submitted material is examined. Therefore, in cases of a strong clinical suspicion of infection, the clinician will initiate prophylactic therapy even in the absence of a positive result on the PFSE.
- If it were to be implemented in the clinical practice, do you consider the costs reasonable? Financial cost and also personnel etc. especially compared to extensive cultures, blood cultures, different culture media etc.
In our hospital, PFSE has already been incorporated into clinical practice through specific operating instruction. This method has been highly successful due to its convenience in terms of cost-effectiveness and the fact that it provides an important result to clinical colleagues in less than thirty minutes of work.
Furthermore, as previously mentioned, it is a test that requires skills that can be easily learned by all medical staff in a short time, including pathologists who do not deal with fetal-placental pathology daily.
Moreover, we inform you that we change the term PFSE with FSMU “frozen section examination of placental membranes and umbilical cord”.
Reviewer 2 Report
Comments and Suggestions for Authors
Few of my observations to improve the manuscript
1. Line 21. This study investigates the potential of frozen section examination (PFSE) of the umbilical cord to improve EONS diagnosis, this sentence looks erroneous. PFSE expansion and Frozen section from placenta or umbilical cord?
2. This is a retrograde study
PFSE enabled risk stratification: high-risk neonates with funisitis received antibi otics, while low-risk neonates with no funisitis often avoided them. How they know about use of antibiotics? They might have used separate criteria.
3. For what purpose the PFSE was done, as this is not related to this study. It looks like part of another study. This type of studies are mostly prospective.
4. Figure 1. (FFPE- Expansion?
5. What is the difference between (PFPR): &PFSE?
6. Out of 55 cases only one has Funisitis by PFSE and 4 FUNISITIS by FFPE slides./PFPR
7. So explain, how should we rely on this by p-value only?
8. As this method of PFSE is not feasible in the most of the centres, justify how it is better than biochemical and clinical markers?
9. Many limitations of the study, including sample size, Retrospective, no comparison group, neonatal outcomes, excess use of antibiotics if any, criteria to start antibiotics with clinical suspicion has has failed, which could have been corrected with PFSE study of the mother.
10. Mention about the ethical clearance from the institute with EC numder.
Author Response
Reviewer: 2
Few of my observations to improve the manuscript
- Line 21. This study investigates the potential of frozen section examination (PFSE) of the umbilical cord to improve EONS diagnosis, this sentence looks erroneous. PFSE expansion and Frozen section from placenta or umbilical cord?
The terminology "PFSE" reflects a more general concept without delving into anatomical details, yet it is still easy to use in clinical communication. However, the text clarifies that it refers to a test performed on amniotic membranes and umbilical cord, and not on the chorionic disc.
Funisitis and chorioamnionitis are closely related: firstly, maternal inflammatory response develops with the development of chorioamnionitis, and subsequently, if the infection persists, fetal inflammatory response is also triggered. Therefore, it would not be prudent to limit the histological investigation to the umbilical cord alone.
However, we agree that the abbreviation PFSE was not consistent with the examination performed. Therefore, we corrected it with “frozen section examination of placental membranes and umbilical cord (FSMU)”
2) This is a retrograde study
PFSE enabled risk stratification: high-risk neonates with funisitis received antibiotics, while low-risk neonates with no funisitis often avoided them. How they know about use of antibiotics? They might have used separate criteria.
The decision to administer antibiotics to the newborn, as outlined in the text, does not depend solely on the outcome of the PFSE. Neonatal sepsis is a worsening event that occurs early (within 48 hours of birth) and can rapidly lead to death or irreversible damage to the newborn. Therefore, in case of significant symptoms in the newborn, the neonatologist will start antibiotic therapy as soon as possible, even before requesting the PFSE. However, the PFSE may still play a role afterward in terms of modulating antibiotic therapy.
3) For what purpose the PFSE was done, as this is not related to this study. It looks like part of another study. This type of studies are mostly prospective
As indicated in the text, the purpose of this retrospective study was primarily to assess the agreement between the results obtained from the frozen section and those obtained from definitive histological examination. This was done to confirm the accuracy of PFSE for inflammatory detection. Indeed, for the risk stratification purpose and therefore in predicting clinical outcomes, the frozen section cannot be considered a unique actor, as it is essential to consider all the other clinical parameters mentioned table 1.
4) Figure 1. (FFPE- Expansion?
Now figure 2
Figure 2 depicts severe chorioamnionitis (A), characterized by widespread neutrophilic granulocyte infiltrates in the chorion with extension into the subamniotic connective tissue. Additionally, severe funisitis (B) is observed with numerous neutrophilic granulocytes in the wall of the umbilical vein.
5) What is the difference between (PFPR): &PFSE?
PFSE (now FSMU) is a method for assessing the fresh placenta with frozen sections, to providing clinicians with information quickly. The PFPR (now PFR) is the definitive and comprehensive placental report, including the umbilical cord, membranes, and disc, which can be obtained after a few days (after formalin-fixation and paraffin embedded).
6) Out of 55 cases only one has Funisitis by PFSE and 4 FUNISITIS by FFPE slides./PFPR. So explain, how should we rely on this by p-value only?
7) So explain, how should we rely on this p-value only?
In our consecutive case series, the occurrences of funisitis were relatively few. Nevertheless, all cases of funisitis were detected by frozen section. This indicates that even though it is a rapid examination, it is also reliable with a very low risk of false negatives. For the single discordant case where funisitis was not confirmed on definitive sections (possible false positive), it should be noted that the frozen section exam can still be affected by artifacts, always emphasizing the importance of close correlation with clinical data.
- As this method of PFSE is not feasible in the most of the centres, justify how it is better than biochemical and clinical markers?
The signs and symptoms of neonatal sepsis are often insidious and nonspecific. Additionally, although blood cultures are still considered the gold standard for diagnosis, they have several limitations. Firstly, they are laboratory tests that require prolonged technical times (48-72 h). Secondly, they have poor sensitivity due to multiple factors, including the small volume of blood that can be sampled, intermittent bacteremia, and frequent administration of intrapartum antibiotic prophylaxis. Furthermore, blood concentrations of inflammation markers are not an effective diagnostic tool since there are no universal reference values, and monitoring involves invasive and traumatic maneuvers for the newborn.
Given these considerations, our test can be a valuable complementary tool due to its low material costs and time requirements, as well as its feasibility for implementation in all laboratories following minimal training.
- Many limitations of the study, including sample size, Retrospective, no comparison group, neonatal outcomes, excess use of antibiotics if any, criteria to start antibiotics with clinical suspicion has has failed, which could have been corrected with PFSE study of the mother.
We are aware of the limitations of our study, particularly concerning the small number of cases analyzed. The lack of a case/control comparison is because both the frozen section examination and the final examination on FFPE samples are performed on the same cases simultaneously.
We have repeatedly emphasized that the frozen section examination should be considered an additional, not exclusive, tool for the clinician in deciding on antibiotic therapy.
For example, in cases clinically of highly suspected sepsis, the clinician will initiate antibiotic therapy regardless of a negative FSMU for funisitis. Conversely, in the absence of significant symptoms but in the presence of funisitis (a marker of fetal inflammatory response), the clinician may choose to start antibiotic therapy.
- Mention about the ethical clearance from the institute with EC number.
The study was conducted according to the guidelines of the Declaration of Helsinki. Ethics committee approval was not required due to the following considerations: all data related to patients’ identification were anonymized; only original slides stained were reassessed and no new sections were produced; this study has a speculative aim, and results will not modify in any way the diagnosis and prognosis or add new clinical information useful for patient management.
Moreover, FSMU is approved in our Standard Operating Procedures (n° IODGN_0008; 08/04/2021).
Moreover, we inform you that we change the term PFSE with FSMU “frozen section examination of placental membranes and umbilical cord”.
Reviewer 3 Report
Comments and Suggestions for Authors
This article compared capacity of frozen section to paraffin-embedded sections through the placental components, to determine funisitis or chorioamniotitis. Interesting topic. It requires major revisions in order to be accepted for publishing:
-in lines 21-22, in Abstract, you wrote: “ This study investigates the potential of frozen section examination (PFSE) of the umbilical cord to improve EONS diagnosis“. First P in PFSE is the first letter of “placental”, as you stated below, and here it suggests the word “potential”. Please rephrase. Second, you also studied membranes, not only umbilical cord sections, so that please add this, too.
- in same Abstract, you wrote:” Additionally, PFSE's role in risk stratification and targeted antibiotic use was explored. PFSE demonstrated high concordance (Kappa=0.88) for funisitis diagnosis, with excellent accuracy (98.3%) and minimal false positives. Moderate concordance was observed for chorioamnionitis stage and grade. PFSE enabled risk stratification: high-risk neonates with funisitis received antibiotics, while low-risk neonates with no funisitis often avoided them. Compared to low-risk infants, the high-risk group had a significantly increased need for antibiotics (Relative Risk: 4.82).“
No. You did discuss about this, a lot, but you did NOT PROVE IT, YOU WROTE NO NUMBER about PFSE in the Tables you presented, and this is the most important negative issue.
I explain below:
-in “Table 3. Concordance between PFSE and PFPR” you wrote:
Table 3. Concordance between PFSE and PFPR. 195
|
196 FUNISITIS Frozen sections |
FUNISITIS FFPE slides |
||||||
|
NO |
S1 |
S2 |
TOTAL |
||||
|
NO |
54 |
0 |
0 |
54 (91.5%) |
|||
|
YES |
1 |
3 |
1 |
5 (8.5%) |
|||
|
TOTAL |
55 (93.2%) |
3 (5.1%) |
1 (1.7%) |
59 (100%) |
|||
|
SIGNIFICANCE LEVEL |
p<0,0001 |
||||||
|
95% CI |
0,64801 to 1 |
||||||
|
CHORIONAMIONITIS STAGE Frozen sections |
CHORIONAMNONTIS STAGE FFPE slides |
||||||
|
NO |
S1 |
S2 |
TOTAL |
||||
|
NO |
38 |
5 |
0 |
43 (72.9%) |
|||
|
S1 |
4 |
7 |
0 |
11 (18.6%) |
|||
|
S2 |
0 |
1 |
4 |
5 (8.5%) |
|||
|
TOTAL |
42 (71.2%) |
13 (22%) |
4 (6.8%) |
59 (100%) |
|||
|
SIGNIFICANCE LEVEL |
p<0,0001 |
||||||
Please watch carefully and notice that in the column “FUNISITIS Frozen sections” YOU WROTE NO NUMBERS OF CASES WITH NO/YES Funisitis. All the numbers are for the FFPE slides, with how many cases have S1, S2 and total. Nothing about the number in Frozen sections.
The same below, in “CHORIONAMNIOTITIS Frozen sections" YOU WROTE NO NUMBERS OF CASES WITH NO/YES Chorioamniotitis. All the numbers are for the FFPE slides, with how many cases have S1, S2 and total. Nothing about the number in Frozen sections.
How could you calculate the “Concordance between PFSE and PFPR” when you did not show the numbers? What did you compare? Where are the numbers? PLEASE WRITE THE NUMBERS.
The same below, in the same Table 3:
|
CHORIONAMIONITIS GRADE Frozen Sections |
CHORIONAMNONTIS GRADE FFPE slides |
||||||
|
NO |
G1 |
G2 |
TOTAL |
||||
|
NO |
37 |
4 |
0 |
41 |
|||
|
G1 |
5 |
9 |
1 |
15 |
|||
|
G2 |
0 |
2 |
1 |
3 |
|||
|
TOTAL |
42 (71.2%) |
15 (25.4%) |
2 (3.4%) |
59 (100%) |
|||
|
SIGNIFICANCE LEVEL |
p<0,0001 |
||||||
|
95% CI |
0,33121 to 0,77517 |
||||||
You wrote no numbers in Frozen sections columns. Please write the numbers.
-and then, in lines 198-203 , under Table 3, you conclude:” Funisitis: High concordance (Kappa coefficient, K = 0.88) was observed. PFSE demonstrated excellent accuracy (98.3%, 95% CI: 90.91-99.96%) with high sensitivity (100%, 95% CI: 39.76-100%) and specificity (98.18%, 95% CI: 90.28-99.95%). Only one case of funisitis identified by PFSE was not confirmed by PFPR (false positive). Chorioamnionitis: Moderate concordance was observed for both stage (K = 0.55) 202 and grade (K = 0.69) of chorioamnionitis. “ Concordance of what? You showed no numbers in frozen sections cases.
-Then you wrote:” Risk Stratification Based on PFSE:
Based on PFSE results, the population was categorized into three risk classes for EONS:
High-Risk (HR; n=5): Infants with any stage of funisitis with or without chorioamnionitis (8.47%). These infants received empiric antibiotic therapy.
Intermediate-Risk (IR; n=1): Infants with no funisitis but with chorioamnionitis Stage 2 (1.69%). These infants were monitored clinically without initial antibiotic treatment.
Low-Risk (LR; n=53): Infants with no funisitis and no chorioamnionitis or chorioamnionitis Stage 1 (89.83%).“
Please explain how you stratified them in high or low risk “based on PFSE results” . It is not clear at all.
-in line 215 you wrote:” Sixteen newborns (27.12%) required empiric antibiotic therapy, all of whom belonged to the HR group”. As you stated above, you only had n=5 infants with high risk, and OUT OF 5, there were 16 who required antibiotic therapy? Please explain.
-in Materials and Methods, please explain why these infants received frozen section examination, and all the other ones did not. How were they selected.
-you compared two methods: frozen and paraffin-embedded. Ok. Still, in Figure 1, you showed only images of paraffin-embedded tissues, which we presumably know already, and no image of frozen section tissue, which is the subject of your original work. That would have been very interesting, to see images of BOTH techniques side by side. Please add at least one image of frozen section.
-at the bottom of Table 2, you showed:

Ok.
Then, below it, you wrote:
“Most patients (72.88%) had absolute indications for PFSE, with PROM being the 187 most frequent (59.32%).“ How did you get 72.88%? In the table you showed, we can only add 59.32+20.34= 79.66% , as the easiest possibility. How did you get that 72.88%?
Then, below, you wrote:” Minor indications included meconium-stained amniotic fluid, pre-eclampsia, gestational diabetes, and cardiotocographic changes (CTG).” In your table, there are “relative” indication, there are no “minor” indications. Is this the same? Please explain/rephrase.
-In References, most titles are old, before 2019 (16 out of 25), please replace some of them with some newer ones.
Author Response
Review 3
This article compared capacity of frozen section to paraffin-embedded sections through the placental components, to determine funisitis or chorioamniotitis. Interesting topic. It requires major revisions in order to be accepted for publishing:
1) in lines 21-22, in Abstract, you wrote: “ This study investigates the potential of frozen section examination (PFSE) of the umbilical cord to improve EONS diagnosis“. First P in PFSE is the first letter of “placental”, as you stated below, and here it suggests the word “potential”. Please rephrase. Second, you also studied membranes, not only umbilical cord sections, so that please add this, too.
We agree that the abbreviation was not consistent with the examination performed. Therefore, we corrected it with “frozen section examination of placental membranes and umbilical cord (FSMU)”
2) in same Abstract, you wrote:” Additionally, PFSE's role in risk stratification and targeted antibiotic use was explored. PFSE demonstrated high concordance (Kappa=0.88) for funisitis diagnosis, with excellent accuracy (98.3%) and minimal false positives. Moderate concordance was observed for chorioamnionitis stage and grade. PFSE enabled risk stratification: high-risk neonates with funisitis received antibiotics, while low-risk neonates with no funisitis often avoided them. Compared to low-risk infants, the high-risk group had a significantly increased need for antibiotics (Relative Risk: 4.82).“
No. You did discuss about this, a lot, but you did NOT PROVE IT, YOU WROTE NO NUMBER about PFSE in the Tables you presented, and this is the most important negative issue.
I explain below:
-in “Table 3. Concordance between PFSE and PFPR” you wrote:
Table 3. Concordance between PFSE and PFPR. 195
Please watch carefully and notice that in the column “FUNISITIS Frozen sections” YOU WROTE NO NUMBERS OF CASES WITH NO/YES Funisitis. All the numbers are for the FFPE slides, with how many cases have S1, S2 and total. Nothing about the number in Frozen sections.
The same below, in “CHORIONAMNIOTITIS Frozen sections" YOU WROTE NO NUMBERS OF CASES WITH NO/YES Chorioamniotitis. All the numbers are for the FFPE slides, with how many cases have S1, S2 and total. Nothing about the number in Frozen sections.
How could you calculate the “Concordance between PFSE and PFPR” when you did not show the numbers? What did you compare? Where are the numbers? PLEASE WRITE THE NUMBERS.
The same below, in the same Table 3: CHORIONAMIONITIS GRADE
You wrote no numbers in Frozen sections columns. Please write the numbers.
and then, in lines 198-203 , under Table 3, you conclude:” Funisitis: High concordance (Kappa coefficient, K = 0.88) was observed. PFSE demonstrated excellent accuracy (98.3%, 95% CI: 90.91-99.96%) with high sensitivity (100%, 95% CI: 39.76-100%) and specificity (98.18%, 95% CI: 90.28-99.95%). Only one case of funisitis identified by PFSE was not confirmed by PFPR (false positive). Chorioamnionitis: Moderate concordance was observed for both stage (K = 0.55) 202 and grade (K = 0.69) of chorioamnionitis. “ Concordance of what? You showed no numbers in frozen sections cases.
I am afraid there was an issue with the formatting of the table that was sent. The numbers are actually all there. Let's try to better explain how Table 3 should be interpreted.
In the fifth column, the numbers related to the freezer examination (FSMU) are reported. Meanwhile, in columns 2, 3, and 4, the numbers related to the final pathological report (FPR) are shown. This table is intended to visualize the level of agreement between the two methods (and to calculate the k coefficient using MedCalc).
For example (funisitis): the frozen examination showed 54 cases (fifth column) with funisitis and 5 cases (fifth column) without funisitis. Of these 5 cases (fifth column), in one case (second column) the presence of funisitis was not confirmed in the final histological examination (FPR).
3) Then you wrote:” Risk Stratification Based on PFSE:
Based on PFSE results, the population was categorized into three risk classes for EONS:
High-Risk (HR; n=5): Infants with any stage of funisitis with or without chorioamnionitis (8.47%). These infants received empiric antibiotic therapy.
Intermediate-Risk (IR; n=1): Infants with no funisitis but with chorioamnionitis Stage 2 (1.69%). These infants were monitored clinically without initial antibiotic treatment.
Low-Risk (LR; n=53): Infants with no funisitis and no chorioamnionitis or chorioamnionitis Stage 1 (89.83%).“
Please explain how you stratified them in high or low risk “based on PFSE results” . It is not clear at all.
FSMU gives a direct indication about the maternal inflammatory response (chorioamnionitis) and fetal response (funisitis). As highlighted by Kovacs et al. (ref 20), funisitis appears to pose a greater risk of fetal complications compared to chorioamnionitis alone. For these reasons, the neonatologist in our institution requests the FSMU in cases where there are risk criteria for EONS. As stated in the text, the FSMU does not have absolute value but should be considered as an additional tool available to the clinician. In these terms, the FSMU can assist the clinician in stratifying the risk of EONS. That said, we agree that our study does not have enough cases (especially a few cases of funisitis) to significantly prove precise risk stratification, and therefore we have removed that paragraph from the text. Indeed, our main objective is to demonstrate the reliability and ease of implementation of the FSMU in clinical practice without implying that this examination should replace other clinical and laboratory investigations for determining the risk of EONS.
4) in line 215 you wrote:” Sixteen newborns (27.12%) required empiric antibiotic therapy, all of whom belonged to the HR group”. As you stated above, you only had n=5 infants with high risk, and OUT OF 5, there were 16 who required antibiotic therapy? Please explain.
We agree with you and we have explain better that sentence.
5) in Materials and Methods, please explain why these infants received frozen section examination, and all the other ones did not. How were they selected.
All 59 cases in our series underwent the frozen section examination. These 59 cases, as noted in the text, are consecutive cases of neonates with risk factors for EONS (period 2019-2021) for whom the freezer examination was requested as a guide for choosing antibiotic therapy.
To better explain ourselves, we have condensed the inclusion and exclusion criteria in the text
6) you compared two methods: frozen and paraffin-embedded. Ok. Still, in Figure 1, you showed only images of paraffin-embedded tissues, which we presumably know already, and no image of frozen section tissue, which is the subject of your original work. That would have been very interesting, to see images of BOTH techniques side by side. Please add at least one image of frozen section.
We follow your suggestion adding a panel of FSMU section (Fig1)
7) at the bottom of Table 2, you showed:
Ok.
Then below it, you wrote:
“Most patients (72.88%) had absolute indications for PFSE, with PROM being the 187 most frequent (59.32%).“ How did you get 72.88%? In the table you showed, we can only add 59.32+20.34= 79.66% , as the easiest possibility. How did you get that 72.88%?
Then, below, you wrote:” Minor indications included meconium-stained amniotic fluid, pre-eclampsia, gestational diabetes, and cardiotocographic changes (CTG).” In your table, there are “relative” indication, there are no “minor” indications. Is this the same? Please explain/rephrase.
The sum of the absolute criteria does not equal 72.88% because, as stated in the text, some cases had more than one major criterion. Specifically, 43 cases (72.88%) had absolute criteria, while 16 cases (27.12%) had only relative criteria. We have added the numbers in the text.
Moreover, we correct “minor” with “relative”
8) In References, most titles are old, before 2019 (16 out of 25), please replace some of them with some newer ones.
We have changed with the following most recent references:
- Boureka E, Krasias D, Tsakiridis I, Karathanasi AM, Mamopoulos A, Athanasiadis A, Dagklis T. Prevention of Early-Onset Neonatal Group B Streptococcal Disease: A Comprehensive Review of Major Guidelines. Obstet Gynecol Surv. 2023 Dec;78(12):766-774. doi: 10.1097/OGX.0000000000001223.
- Kovács K, Kovács ŐZ, Bajzát D, Imrei M, Nagy R, Németh D, Kói T, Szabó M, Fintha A, Hegyi P, Garami M, Gasparics Á. The histologic fetal inflammatory response and neonatal outcomes: systematic review and meta-analysis. Am J Obstet Gynecol. 2024 May;230(5):493-511.e3. doi: 10.1016/j.ajog.2023.11.1223.
Reviewer 4 Report
Comments and Suggestions for Authors
The authors reported to use of frozen section examination of umbilical cord to identify funisitis as an indicator of Neonatal sepsis, and concluded it has a high accuracy in detecting funisitis. This in turn will prevent unnecessary antibiotic usage and prevent neonatal microbiome dysbiosis.
Here are my comments:
The introduction is too lengthy, suggest to concise it. Also, some of the paragraphs consist of only 1 or 2 sentences. They are too short to form a paragraph. Suggest to combine them into a longer paragraph.
Figure 1: What is FFPE? All abbreviation needs to be clear stated.
Figure 1: Spelling of chorioamnionitis is incorrect. Also, what do you mean severe funisitis? How is funisitis grade and stage? What is the definition of severe in this context?
The authors should include figures of the frozen section tissue samples such as umbilical cord and membrane too. The quality of frozen section preparation plays a crucial role in this study.
Methodology:
The number of cases and control should be included in the methodology.
Describe which ethics committee approved the study and provide the ethics approval code.
Page 184: Why the need to pair male and female babies in this study?
Three sections were taken from the umbilical cord. Where were the 3 sections taken? Near the placenta? near the fetus or at the mid-point?
Where was the membrane taken? Near umbilical cord insertion site or away?
How was the placenta sampled for the FFPE tissue blocks and analysis?
Page 146, define somministrated.
Exclusion criteria: ….without the PFSE, should not be considered as an exclusion criteria.
Why toluidine blue stain was performed on the frozen section?
Table 3:
Explain what are S1 and S2.
What is Funisitis: HiStrong?
The table 3 presentation is not clear. I don’t understand how frozen section was being compared with FFPE findings. Frozen is in column one? How many cases have funisitis in the frozen section samples?
The total number of cases were too small to make a good conclusion. In particularly, there were only 5 cases of funisitis.
The conclusion should focus on the findings of frozen section, rather than on the neonatal microbiome, which is not part of this study. Whether or not neonates will have a good microbiome eventually was not a finding in this study. Discussion should also focus on the need of performing frozen section on suspected chorioamnionitis and funisitis.

None
Author Response
Reviewer: 4
The authors reported to use of frozen section examination of umbilical cord to identify funisitis as an indicator of Neonatal sepsis, and concluded it has a high accuracy in detecting funisitis. This in turn will prevent unnecessary antibiotic usage and prevent neonatal microbiome dysbiosis.
Here are my comments:
1) The introduction is too lengthy, suggest to concise it. Also, some of the paragraphs consist of only 1 or 2 sentences. They are too short to form a paragraph. Suggest to combine them into a longer paragraph.
In the introduction, we tried to provide as much context as possible, resulting in it being somewhat lengthy. Following your advice, we have made efforts to condense the material and simplify the text to make it more manageable.
2) Figure 1: What is FFPE? All abbreviation needs to be clear stated.
Figure 1: Spelling of chorioamnionitis is incorrect.
"Formalin-fixed paraffin-embedded" (FFPE) refers to tissue samples that have been fixed in formalin, embedded in paraffin, and then sectioned for histological analysis. Therefore, we have added the explanation of the abbreviation in the text.
Furthermore, we have corrected the spelling of the term chorioamnionitis.
3) What do you mean severe funisitis? How is funisitis grade and stage? What is the definition of severe in this context?
According to the cited Amsterdam Consensus chorioamnionitis is staged according to the following criteria:
Stage 1: neutrophils in chorion laeve of the extraplacental membranes (chorionitis)
Stage 2: neutrophils within chorionic or sub-amniotic connective tissue
Stage 3: stage 2, plus necrosis of amnionic epithelium
The grading depends on the amount of neutrophils (G1 or G2).
In our case series we found only Stage 1 or 2 cases.
Funisitis is staged according to the following criteria:
Stage 1: fetal neutrophils within chorionic plate vessel walls (fetal vasculitis) or umbilical vein vessel wall (umbilical vein vasculitis)
Stage 2: fetal neutrophils within umbilical arteries (umbilical artery vasculitis) or vein
Stage 3: necrotizing funisitis (with dense neutrophilic aggregates into Warthon jelly)
The grading depends on the amount of neutrophils (G1 or G2).
In our case series we found only Stage 1 or 2 cases.
In our study, we considered as “severe” the cases with funisitis (stage II and grade 1; stage I and grade 2) and cases of chorioamnionitis (stage II and grade 1; stage I and grade 2).
Therefore, in the text, we have better specified the reference to the Amsterdam criteria (Redline et all criteria) for the correct classification.
4) The authors should include figures of the frozen section tissue samples such as umbilical cord and membrane too. The quality of frozen section preparation plays a crucial role in this study.
Following your advice, we have prepared a panel containing images of chorioamnionitis and funisitis in PFSE.
5) Methodology:
The number of cases and control should be included in the methodology.
In the Materials and Methods section, we explained the nature of the case series (consecutive neonates who underwent PFSE between 2019 and 2021), with the inclusion and exclusion criteria. In the Results section and in the table, we instead reported the numerical values from our case series. Specifically, there were 59 neonates for each of whom both PFSE and PFPR records were available.
6) Describe which ethics committee approved the study and provide the ethics approval code.
The study was conducted according to the guidelines of the Declaration of Helsinki. Ethics committee approval was not required due to the following considerations: all data related to patients’ identification were anonymized; only original slides stained were reassessed and no new sections were produced; this study has a speculative aim, and results will not modify in any way the diagnosis and prognosis or add new clinical information useful for patient management.
Moreover, PFSE is approved in our Standard Operating Procedures (n° IODGN_0008; 08/04/2021).
7) Page 184: Why the need to pair male and female babies in this study?
We have included the sex of the neonates for completeness, but we have not highlighted any statistically significant differences between the two sexes.
8) Three sections were taken from the umbilical cord. Where were the 3 sections taken? Near the placenta? near the fetus or at the mid-point?
Where was the membrane taken? Near umbilical cord insertion site or away?
How was the placenta sampled for the FFPE tissue blocks and analysis?
In the text, we have provided clearer specifications regarding the areas from which we conducted the biopsies for examination in PFSE.
Moreovere we added in the text the following sentence:
“For the conclusive histological examination, three more chorionic disc biopsies were taken (within the central two-thirds of the disc, with one encompassing the umbilical cord insertion area). The two blocks utilized for placental frozen section analysis were thawed and paraffin-embedded for the definitive histological examination”.
9) Page 146, define somministrated.
The study aims to demonstrate the usefulness of PFSE in identifying neonatal sepsis quickly and accurately. Therefore, we haven't delved into the details of antibiotic therapies. However, the therapy of choice consists of a combination of Ampicillin + Gentamicin. In case of clinical worsening after 36 hours of therapy and/or meningitis, a combination with a third-generation cephalosporin (Cefotaxime or Ceftazidime) is recommended. However, their use remains limited as they promote the emergence of multi-resistant Gram-negative strains and systemic infections by Candida. Antibiotic therapy should be discontinued when blood cultures are sterile at 48-72 hours of incubation unless the clinical presentation is particularly suggestive or there is a site-specific infection. In the case of confirmed sepsis, treatment should be continued for 10 days, 14 days if there is GBS bacteremia and 21-28 days in the case of meningitis. Determination of blood levels of aminoglycosides should be performed when therapy with these drugs is continued for more than 72 hours or in the presence of renal insufficiency: if the levels remain below 2 mcg/ml, the risk of ototoxicity and nephrotoxicity is significantly reduced.
10) Exclusion criteria: ….without the PFSE, should not be considered as an exclusion criteria.
The absence of PFSE is inherently an exclusion criterion for the study. As elucidated in the text, it is at the discretion of the clinician to opt for this test depending on the specific requirements of each case. For instance, in scenarios involving multiple sepsis risk factors and potentially noteworthy clinical symptoms in the neonate, the clinician might occasionally administer antibiotic therapy without the help of PFSE. In such instances, the placenta is exclusively dispatched to the pathology laboratory for the definitive examination. Although infrequent, these cases were omitted from the case series due to their inability to facilitate a comparison between the intraoperative examination and the definitive examination which is the main scope of this study.
11) Why toluidine blue stain was performed on the frozen section?
For the frozen section examination, we utilize not only hematoxylin and eosin but also toluidine blue to enhance nuclear staining. Specifically, in the case of PFSE, it aids in better visualization of the segmented nuclei of neutrophilic granulocytes.
12) Table 3:
Explain what are S1 and S2.
“S” stands for stage. Stage 1 and stage 2.
What is Funisitis: HiStrong?
We correct and deleted it
13) The table 3 presentation is not clear. I don’t understand how frozen section was being compared with FFPE findings. Frozen is in column one? How many cases have funisitis in the frozen section samples?
Indeed. This table provides a comparison of cases showing the absence of inflammation, chorioamnionitis, and funisitis between the two groups (PFSE and PPRT). Specifically regarding funisitis, we identified 5 cases during the freezer examination. Among these, 4 were confirmed in the definitive examination, while one turned out to be unconfirmed (false positive) and deepened in discussion.
14) The total number of cases were too small to make a good conclusion. In particularly, there were only 5 cases of funisitis.
We are aware that one of the limitations of the study is the small sample size. However, some strengths include the consecutive nature of the cases, thus representing what occurs in clinical practice. Nevertheless, by combining the data related to chorioamnionitis and considering the total of 54 cases, the good reliability of the freezer examination in excluding the presence of inflammation can be inferred.
In total, the frozen examination failed to detect inflammation in the membranes in only 5 out of 59 cases compared to the definitive examination. However, in these 5 cases, severe chorioamnionitis (all cases S1) was absent, potentially having minimal impact on the newborn's health. Nevertheless, we stress that this examination should be regarded as supplementary and not as a sole replacement for other clinical investigations.
15) The conclusion should focus on the findings of frozen section, rather than on the neonatal microbiome, which is not part of this study. Whether or not neonates will have a good microbiome eventually was not a finding in this study.
Discussion should also focus on the need of performing frozen section on suspected chorioamnionitis and funisitis.
We have modified both discussion and conclusion as you suggested.
Round 2
Reviewer 2 Report
Comments and Suggestions for Authors
Nicely addressed the issues raised.
Author Response
Thank you for your suggestions
Reviewer 3 Report
Comments and Suggestions for Authors
It is ok now. Thank you for the changes you made.
Author Response
Thank you for your suggestions
Reviewer 4 Report
Comments and Suggestions for Authors
Comment no 6 on ethics committee:
As this a prospective study that required an additional section of the umbilical cord for frozen section. It should have an ethics committee approval. Moreover, it may also need to have written consent from the mother. Please provide a authorise letter from your institution that this study does not require ethics approval and state clearly the nature of the study.
Comment no 7 on matched male and female:
The authors did not explain the reason. What do you mean completeness? Do you expect any statistical difference between the 2 groups? If yes, why?
Comment no 8 on sections from UC:
“For the conclusive histological examination, three more chorionic disc biopsies were taken (within the central two-thirds of the disc, with one encompassing the umbilical cord insertion area). The two blocks utilized for placental frozen section analysis were thawed and paraffin-embedded for the definitive histological examination”.
The above paragraph does not describe umbilical cord. All sections are from the placental disc. Please explain.
Further comments: Study has shown that different region of the umbilical cord has different rate of identifying inflammation.
Comment no 10 on exclusion criteria:
The authors do not understand what I meant. If the clinicians did not take the PFSE, then it is due to specific reasons. The reasons are the exclusion criteria.
Comment no 12 on table 3:
S as stage need to be in the footnote.
Final comment:
My major concern is there are only 5 cases with funisitis, but the aim of this study is identifying early neonatal sepsis. Of the 59 cases, 54 cases did not have funisitis (which is the indicator of fetal response).
If someone ask you, where should I take the placenta for frozen section to identify early neonatal sepsis, what would be your respond?
Comments on the Quality of English LanguageSome of the sentences are not clear. It would be better to have some English editing services.
Author Response
Thank you for your comments and suggestions. Below, we will further explain the following points.
1) Comment no 6 on ethics committee:
As this a prospective study that required an additional section of the umbilical cord for frozen section. It should have an ethics committee approval. Moreover, it may also need to have written consent from the mother. Please provide a authorise letter from your institution that this study does not require ethics approval and state clearly the nature of the study.
We will try to explain the concept further. As specified in the text (Materials and Methods, line 151), this is a completely retrospective study (not prospective). All the data (clinical and pathological) for the study were collected retrospectively. In this sense, both the frozen section data (FSMU) and the definitive pathological examination data (FPR) were collected retrospectively. Therefore, the samples taken from the umbilical cord and membranes for the FSMU were not taken at later times for the study but were routinely performed by our standard operating procedure.
As previously mentioned, the frozen section examination of membrane and umbilical cord is included as a procedure within the standard operating instructions of our Institution (last revision no. IODGN_0008; 08/04/2021). These procedures are directly reviewed by our healthcare management and the Quality control authority. For these reasons, there are no further ethical requirements concerning this retrospective study.
Additionally, in our institution, no specific informed consent is required for the cryostat examination, just as it is not required for the definitive pathological examination.
2) Comment no 7 on matched male and female:
The authors did not explain the reason. What do you mean completeness? Do you expect any statistical difference between the 2 groups? If yes, why?
This data was reported in Table 2 solely for epidemiological purposes. It was not our intention to investigate any difference on the subject between males and females.
3) Comment no 8 on sections from UC:
“For the conclusive histological examination, three more chorionic disc biopsies were taken (within the central two-thirds of the disc, with one encompassing the umbilical cord insertion area). The two blocks utilized for placental frozen section analysis were thawed and paraffin-embedded for the definitive histological examination”.
The above paragraph does not describe umbilical cord. All sections are from the placental disc. Please explain.
Further comments: Study has shown that different region of the umbilical cord has different rate of identifying inflammation.
You are right. The sentence was unclear. We have added the following sentence to the text (line 124).
“In the same session, three chorionic disc samples were taken (within the central two-thirds of the disc, with one encompassing the umbilical cord insertion area) for subsequent FPR. The two frozen blocks utilized for FSMU analysis were then thawed for FPR, which includes fixation in 10% buffered formalin for 12 hours, processing, paraffin embedding (FFPE), sectioning at 3 μm thickness, and staining with H&E”.
Furthermore, our study did not demonstrate a difference based on the site of the cord sampling.
4) Comment no 10 on exclusion criteria:
The authors do not understand what I meant. If the clinicians did not take the PFSE, then it is due to specific reasons. The reasons are the exclusion criteria.
We noticed that the following sentence “This study excluded neonates without Table 1 risk factors, or the antibiotic therapy was administered before FSMU evaluation or FSMU report was not available” was repetitive, so we removed it from the text. For this case series, we considered all cases in which FSMU was requested during the specified period (2019-2021). As specified in the text, this examination is requested in our institution (according to the specific standard operating procedure) by the clinician as an additional test for evaluating the risk of EONS, in the presence of the risk factors identified in Table 1. Therefore, all the cases in our study underwent both the FSMU and the FPR examinations, as per the protocol.
It would not have made sense for the clinician to request a frozen section examination to search for inflammation of the membranes and umbilical cord if there were no risk factors for EONS.
5) Comment no 12 on table 3:
S as stage need to be in the footnote.
You are right. We preferred to write out the word 'Stage' instead of using the abbreviation 'S'.
So we delete “S” from the text.
6) Final comment:
My major concern is there are only 5 cases with funisitis, but the aim of this study is identifying early neonatal sepsis. Of the 59 cases, 54 cases did not have funisitis (which is the indicator of fetal response).
If someone ask you, where should I take the placenta for frozen section to identify early neonatal sepsis, what would be your respond?
The main aim of our study is to retrospectively evaluate the concordance between the results of the frozen section examination performed on the membranes and umbilical cord, compared to the definitive pathological examination. As we reported at the beginning of the discussion (line 192) the incidence of neonatal sepsis is fortunately relatively low (0.61/thousand live births). This data does not appear to disagree with the data of our cases, in which of the 59 consecutive cases, despite the presence of risk factors for EONS, less than 10 percent of the cases had morphological signs of fetal inflammatory response (funisitis).
Furthermore, the aim of our study was not to demonstrate the superiority of the frozen section examination compared to other clinical-laboratory investigations accessible to the clinician. Our objective was to show good reliability with a short turnaround time, which is therefore a useful diagnostic tool within an integrated diagnostic path for the prevention of EONS.
To the question of where to take the sample, the Literature indicates funisitis as an indicator of fetal inflammatory response. Although we are aware of the limitations linked to the small number of cases, in our study no cases of "true" funisitis (4 cases) were missed during FSMU. Therefore, to exclude the presence of funisitis, the frozen section examination of the umbilical cord appears to be a useful diagnostic tool in routine practice.